# The Impact of Genetics on Gut Microbiota of Growing and Fattening Pigs under Moderate N Restriction

**DOI:** 10.3390/ani11102846

**Published:** 2021-09-29

**Authors:** Laura Sarri, Sandra Costa-Roura, Joaquim Balcells, Ahmad Reza Seradj, Gabriel de la Fuente

**Affiliations:** Departament de Ciència Animal, Universitat de Lleida-Agrotecnio-CERCA Center, Av. Alcalde Rovira Roure 191, 25198 Lleida, Spain; laura.sarri@udl.cat (L.S.); sandra.costa@udl.cat (S.C.-R.); joaquim.balcells@udl.cat (J.B.); reza.seradj@udl.cat (A.R.S.)

**Keywords:** microbiota, swine, intestinal tract, protein restriction, pig producing type

## Abstract

**Simple Summary:**

The microbial population, diversity and interactions along the intestinal tract (including ileum, cecum and distal colon) were assessed in two producing types of pigs, where castrated Duroc pigs were used as heavy pigs, whereas entire F2 crossbred pigs (Pietrain sires × (F1: Duroc × Landrace) dams) were used as lean ones. Half of the animals belonged to two production phases (growing vs. fattening) and were subjected to a moderate crude protein restriction (2%). The producing type of pig and the production phase were the effects that most affected the parameters studied, where the fattening pigs and the lean ones showed higher alpha diversity indices and microbial network complexity. However, the lower dietary crude protein content had only a slight effect on microbial networks. These findings provide further understanding about how different effects (production phase, producing type, dietary crude protein level and intestinal segment) interact and affect gut microbiome, which could be taken into account for the optimization of pork production efficiency.

**Abstract:**

Characterization of intestinal microbiota is of great interest due to its relevant impact on growth, feed efficiency and pig carcass quality. Microbial composition shifts along the gut, but it also depends on the host (i.e., age, genetic background), diet composition and environmental conditions. To simultaneously study the effects of producing type (PT), production phase (PP) and dietary crude protein (CP) content on microbial populations, 20 Duroc pigs and 16 crossbred pigs (F2), belonging to growing and fattening phases, were used. Half of the pigs of each PT were fed a moderate CP restriction (2%). After sacrifice, contents of ileum, cecum and distal colon were collected for sequencing procedure. Fattening pigs presented higher microbial richness than growing pigs because of higher maturity and stability of the community. The F2 pigs showed higher bacterial alpha diversity and microbial network complexity (cecum and colon), especially in the fattening phase, while Duroc pigs tended to have higher Firmicutes/Bacteroidetes ratio in cecum segment. *Lactobacillus* was the predominant genus, and along with *Streptococcus* and *Clostridium*, their relative abundance decreased throughout the intestine. Although low CP diet did not alter the microbial diversity, it increased interaction network complexity. These results have revealed that the moderate CP restriction had lower impact on intestinal microbiota than PP and PT of pigs.

## 1. Introduction

Optimizing feed efficiency is one of the current challenges for the swine industry that also promotes the reduction of both feeding costs and environmental impact [1,2]. Within this framework, reducing crude protein (CP) level in diets balanced with synthetic amino acids is a widely used strategy which allows the improvement of nitrogen utilization and the reduction of the nitrogen load from manure [3]. This approach complies with the European Union Directive 91/676/EEC concerning the protection of waters against nitrates [4]. In addition, the growing demand for pork products of improved palatability and nutritional value has led to the development of alternative production systems of great economic importance [5] based on breeds with specific fatness traits to produce dry-cured ham or premium pieces, such as Duroc or Iberian pigs.

Together with management (nutrition, biosecurity, vaccination) and genetic plans, implementation of new production strategies should also consider the animal’s physiology [2,6]. In this regard, it has been evidenced that gut microbiota plays important roles in promoting immune system development [7], regulating host nutrient metabolism [8,9], modulating phenotypic traits [9,10] and producing beneficial substances [11]. Gut microbiota consists of a complex ecosystem that is established through a sequence of dynamic successions of the dominant microbial groups. These changes occur throughout the entire intestinal tract and over time [12] in order to adapt to endogenous and exogenous factors to which the individual is subjected. Substantial progress in high-throughput sequencing techniques have enabled the characterization of gut microbial communities and their interaction with the host, which has received increasing interest in recent decades due to its potential contribution to production traits.

Since there is high potential to modulate gut microbiome in different managing scenarios [13], it would be possible to reshape the community structure to achieve different productive goals [9]. For this purpose, a further understanding of the factors affecting gut microbiota and their interactions seems essential. Thus, the objective of the present study was to evaluate the effects of a moderate CP restriction and the producing type (PT) of pigs (Duroc pigs as heavy and F2 crossbreed as lean pigs) on gut microbiome structure across the intestinal tract (ileum, cecum and distal colon) throughout the animal’s development (growing and fattening phases).

## 2. Materials and Methods

Protocols and experimental procedures were approved by the Ethics Committee for Animal Experimentation of the University of Lleida, under Project License CEEA 09-05/16. Care and use of animals were in accordance with the Spanish Policy for Animal Protection RD 53/2013, which meets the European Union Directive 2010/63 on the protection of animals used for experimental and other scientific purposes.

### 2.1. Animals, Diets and Sampling Procedure

A total of 36 male pigs belonging to two production phases (PPs; growing and fattening), were housed in the same environmentally controlled room at the Swine Research Center located in Torrelameu (CEP; Lleida, Spain). Twenty of these 36 pigs were surgically castrated purebred Duroc pigs, of which 12 were in the growing phase (mean ± standard error: 26.4 ± 3.11 kg of body weight (BW) at sacrifice) and 8 in the fattening phase (86.1 ± 2.74 kg BW), while the remaining 16 pigs were entire crossbreds (F2) (Pietrain sires × (F1: Duroc × Landrace) dams), 8 in the growing phase (30.5 ± 1.36 kg BW) and 8 in the fattening phase (91.1 ± 1.23 kg BW). Moreover, two experimental diets were formulated for each PP, which were provided ad libitum for 15 days (4 days of dietary changeover plus 11 days of adaptation) prior to slaughter. Both diets were isoenergetic and identical in covering the nutritional requirements but differing in 2% of CP concentration. Half of the pigs of each PT were allotted to one of the two dietary treatments, standard protein (SP) diet and low protein (LP) diet, formulated with 17% and 15% CP in the growing phase and 15% and 13% CP in the fattening phase, respectively. All diets were supplemented with synthetic amino acids to meet the nutrient requirements recommended by FEDNA [14]. The ingredients and chemical composition of the diets are provided in Table 1. The ambient temperature during the entire experimental period was maintained between 23.5 and 25 °C, under a mechanical ventilation system and natural light. Moreover, animals of the same PP and PT were placed in groups of four in 55% concrete slatted-floor pens (2.1 × 2 m^2^) for the first 9 days, and then placed individually in metabolic cages during the last 6 days, as was previously described in [15].

On the last day of the trial, pigs were sacrificed, and the contents of ileum, cecum and distal colon were collected simultaneously for microbial characterization. Samples were immediately frozen in dry ice and stored at −40 °C until further analysis.

### 2.2. Genomic DNA Extraction, 16S rRNA Amplicon Sequencing and Bioinformatics

Microbial genomic DNA was extracted using DNeasy PowerLyzer PowerSoil Kit (Qiagen, Hilden, Germany) following the manufacturer’s instructions. Mock community DNA was included as positive control for library preparation (Zymobiomics Microbial Community DNA, ZymoResearch, Irvine, CA, USA).

Samples were amplified using primers 341F and 805R, which target the V3–V4 region of the bacterial and archaeal 16S rRNA. PCR was performed in 10 μL final volume with 0.2 μM primer concentration. The PCR included: 3 min at 95 °C (initial denaturation) followed by 25 cycles of 30 s at 95 °C, 30 s at 55 °C and 30 s at 72 °C, and a final elongation step of 5 min at 72 °C. PCR products were purified using AMPure XP beads (Beckman Coulter, Nyon, Switzerland) with a 0.9× ratio according to the manufacturer’s instructions.

The paired-end sequencing was conducted following an Illumina Miseq sequencing 300 × 2 approach. Quality control filtering and OTU binning of the resulting sequences were executed using DADA2 software [16]. Finally, taxonomic assignment of phylotypes was performed using a Bayesian classifier trained with Silva database [17]. Extraction and sequencing of DNA and bioinformatic procedures were carried out by Microomics Systems, S.L. (Barcelona, Spain).

### 2.3. Statistical Analysis

Analysis described below were performed in duplicate aiming to assess: (i) differences in ileal, cecal and colonic microbiota between PTs (Duroc vs. F2) in both PPs; and (ii) differences in ileal, cecal and colonic microbiota between experimental diets (SP vs. LP) in both PPs.

Sequence data were normalized to the same mean and alpha diversity indices were calculated (R Core Team, 2020; vegan package) to measure the variability of species within a sample. Data were then analyzed with a linear model including PT × PP and diet × PP as fixed effects (R Core Team, 2020; stats package). Data from the three intestinal segments (ileum, cecum and distal colon) were analyzed separately to decrease datasets sparsity (certain OTUs were not present in each one of the three locations and, therefore, working on a single dataset would highly increase the number of zeros). Contrasts between either PT or diets were performed by Tukey’s test (R Core Team, 2020; emmeans package). Individual samples out of three standard deviations of the mean were discarded and not included to the statistical analysis. Significant effects were declared at *p* < 0.05 and tendency to difference at *p* between 0.05 and 0.10.

To circumvent the compositional bias problem [18,19], we applied the Aitchison’s centered log ratio (clr) transformation to carry the data to a Euclidean space, after replacing zeros by adding 1 to each value. To measure differences in microbiome composition between samples, beta diversity was approached through performing a partial least squares-discriminant analysis (PLS–DA) based on clr (R Core Team, 2020; mixOmics package). To test whether differences in microbiota composition between treatments were statistically significant, a permutational multivariate analysis of variance (PERMANOVA) was conducted based on the clr Euclidean distance, including PT × PP and diet × PP interactions and calculating statistical significance after 10,000 random permutations (R Core Team, 2020; vegan package). To decipher which genera abundance were responsible for the differences between treatments, an ANOVA-like differential expression (ALDEx) analysis was conducted over those genera present at least at 50% of the individuals (R Core Team, 2020; Aldex2 package) [20]. Finally, to describe the interactions within ileum, cecum and colon microbial community, we performed a network analysis through Sparse Correlations for Compositional data (SparCC) technique (R Core Team, 2020; SpiecEasi package) [21] over those genera present in at least 50% of the individuals. Microbial networks were graphically represented (R Core Team, 2020; igraph package) and their complexity was described in terms of number of nodes (genera), number of edges (significant positive or negative correlations), node degree (number of connections that any node establishes with other nodes) and betweenness centrality (measure of centrality in a graph based on shortest paths). Differences in microbial composition between intestinal segments were described by (i) graphical representation of the relative abundance of major genera (>1% analyzed sequences); (ii) graphical representation of core microbial community in each intestinal segment; (iii) alpha diversity indices, statistically assessed using a linear model including segment × PP as fixed effects; and (iv) beta diversity, approached by the graphical representation of PLS–DA and PERMANOVA.

## 3. Results

### 3.1. Dataset Features

Sequencing generated a total of 3,709,916 high-quality sequences obtained from the 108 digestive content samples from the three intestinal regions. The mean number of sequences per sample was 34,251 for the ileum, 31,167 for the cecum and 37,635 for the distal colon. These sequences resulted in a total of 181 OTUs in ileum (33 OTUs per sample), 395 OTUs in cecum (107 OTUs per sample) and 453 OTUs in colon (136 OTUs per sample). The unclassified rate of OTUs at genera level increased along the gut segments, with 8.9%, 11.3% and 14.2% for ileum, cecum and colon samples, respectively.

### 3.2. Alpha Diversity

Microbial diversity indices along with the Firmicutes/Bacteroidetes ratio were analyzed in both PTs of pigs in each PP (Table 2). The Firmicutes/Bacteroidetes ratio is not presented for the ileum segment due to the low abundance of Bacteroidetes in that intestinal segment. This was also analyzed for the dietary CP content in growing and fattening phases (Appendix A), although diet effect was negligible.

The PT had a significant influence on microbial diversity (Table 2). F2 pigs harbored a more diverse bacterial community compared to Duroc pigs along the intestinal tract. Significant differences were detected in ileum (*p* = 0.014 and 0.009 for Shannon and Simpson indices, respectively), cecum (Shannon index, *p* = 0.033) and colon (Simpson index, *p* = 0.047). Moreover, F2 pigs showed higher microbial richness in the cecum (*p* < 0.010), and higher evenness in both ileum and distal colon segments (*p* = 0.011) compared to Duroc. In all three intestinal segments, fattening pigs presented a significantly higher microbial richness than growing ones (*p* < 0.05). A significant interaction was also found in ileum between PT and PP effects (*p* = 0.014), where fattening F2 pigs had significantly higher microbial richness than growing F2 pigs, although no differences were detected between growing and fattening Duroc pigs.

The Firmicutes/Bacteroidetes ratio between both PTs in growing and fattening phases was only calculated in cecum and distal colon. In the cecum, Duroc pigs showed a slight tendency to have higher ratio than F2 pigs (*p* = 0.089) with no variation with PP. Although no significant differences between PTs nor PPs were obtained in the distal colon, the Firmicutes/Bacteroidetes ratio in Duroc pigs numerically decreased from growing to fattening phases, whereas in F2 pigs it remained constant.

### 3.3. Microbial Composition throughout the Intestinal Tract

Alpha diversity indices were also obtained between intestinal segments in the two PPs (Appendix A), in which the ileum segment had lower Shannon and Simpson indices, microbial richness and evenness than the lower intestine segments (cecum and distal colon) in both growing and fattening phases (*p* < 0.001). In addition, no significant differences between the cecum and distal colon were detected in each PP, except for microbial richness in the growing phase that was significantly higher in the distal colon. This is graphically evidenced in Figure 1 where the relative abundance of the main bacterial genera is presented. The lower intestinal regions exhibited a higher number of genera with higher evenness in their relative abundance than those in the upper intestine. In relation to the PP (Appendix A), when all intestinal segments were considered, fattening pigs showed significantly higher Shannon index and microbial richness than growing pigs (*p* < 0.05), as well as a slight tendency to have higher Simpson index (*p* = 0.084). However, only Simpson index in ileum and microbial richness in the cecum presented significant differences.

In both PPs, the microbial community displayed similar dynamics between Duroc and F2 pigs (Figure 1). *Lactobacillus* was the predominant genus in the ileum of growing and fattening pigs, and it decreased progressively from ileum to distal colon. The same dynamics were observed in *Streptococcus* and *Bifidobacterium* (growing phase, Figure 1A,B), and *Clostridium sensu stricto 1* and *Terrisporobacter* (fattening phase, Figure 1C,D), which seemed to almost exclusively colonize the upper intestinal regions. Many other genera with similar relative abundance were found in the cecum, of which *Megasphaera*, *Eubacterium*, *Dialister* and *Mitsuokella* stood out in the growing phase, maintaining or even increasing their abundance in the distal colon (Figure 1A,B). However, in the fattening phase, *Alloprevotella* was abundant in the cecum of both PTs (Figure 1C,D), and *Methanobrevibacter*, *Lachnospiraceae XPB1014* group, *Ruminococcaceae UCG-02* and *Ruminococcaceae UCG-05* were enriched in the colon of fattening Duroc pigs (Figure 1C).

The core microbiota was identified along the ileum (Appendix A), cecum (Appendix A) and distal colon (Appendix A) as the shared OTUs by all individuals at each PP. The upper intestine showed higher number of shared sequences than the lower gut segments, with 57% and 48% of the sequences in the ileum of growing and fattening pigs, respectively (Appendix A). *Unclassified Lactobacillus* was the predominant shared OTU throughout the intestine. It accounted for more than 71% of the shared sequences in the ileum, and progressively decreased to represent 29.6% in the colon of fattening pigs (Appendix A). The colon of growing pigs was the exception, in which the most shared OTU was *unclassifiedMitsuokella* with 13.8% of the shared sequences (Appendix A).

### 3.4. Beta Diversity

Figure 2 graphically represents the beta diversity of the microbial community inhabiting the three intestinal segments studied in the animals fed the two dietary treatments in growing and fattening phases. The effect of PP was important in all intestinal segments (*p* < 0.001), while no significant differences were found for the dietary CP content (*p* > 0.433), although it seemed to be more evident in the growing phase than in the fattening phase. The beta diversity of the two PTs belonging to both PPs is represented in Figure 3. The overall clustering of microbial community samples suggested that microbiota in the ileum (Figure 3A), cecum (Figure 3B) and distal colon (Figure 3C) was different when growing and fattening animals were compared (*p* < 0.001). In a similar manner, differences in microbial community composition between the two PTs were found in ileum and cecum segments (*p* < 0.05, Figure 3A,B) but disappeared in colon (*p* = 0.171, Figure 3C); nevertheless, statistical differences in genera abundance between PTs could not be detected by ALDEx analysis, regardless of intestinal segment.

The beta diversity analysis of microbial OTUs in ileum, cecum and distal colon is graphically represented in Appendix A, in which the clustering is clearly differentiated among the three intestinal segments (*p* < 0.001), although the microbial communities of the ileum appeared to be more distinct than those of the more distal segments.

### 3.5. Microbial Networks

Microbial networks were built to test interactions among microbial genera in each intestinal segment. Degree of interaction was studied through the number of genera (nodes) that established significant interactions (edges) with other genera, as well as the number of interactions established per node (node degree).

The overall microbial network complexity increased across the intestinal segments: the mean number of nodes and edges in each intestinal segment were 12 and 13 in ileum, 50 and 155 in cecum and 68 and 230 in distal colon (Appendix A). In the ileum segment, the most notable differences in microbial networks were observed between PPs (Appendix A); growing pigs showed more complexity in their microbial networks than fattening pigs, in terms of number of nodes, edges and node degree (Appendix A). Regarding dietary CP content, pigs fed the LP diet showed more complex networks through the intestine than those fed the SP diet, especially in ileum (Appendix A) and colon segments (Appendix A), while in the cecum (Appendix A) it was only detected in the growing phase. In turn, F2 pigs exhibited more complex microbial networks than Duroc pigs in cecum (Appendix A) and distal colon (Appendix A), in both growing and fattening phases. However, a different evolution in microbial network complexity in cecum and colon segments was detected between the two PTs tested; while microbial network complexity increased considerably as animals aged in the case of F2 pigs, in Duroc pigs it remained constant in both PPs.

Betweenness centrality, which measures the possible influence of an individual node on other nodes, was also calculated (Appendix A). This measure was higher in growing LP-fed pigs, especially in the colon and the cecum segments, and in the cecum of fattening SP-fed pigs. Moreover, it was higher in the microbial network of F2 pigs compared to Duroc ones and tended to increase from growing to fattening phases in both PTs, with the sole exception of the ileum of Duroc pigs. The increase of betweenness centrality with age was more pronounced in F2 pigs.

## 4. Discussion

Authors are aware of the limiting number of animals per group with the 2 × 2 × 2 factorial design to reach consistency in both results and conclusions. The present study was part of a complex trial in which animals’ protein and lipid metabolisms were also studied [15]. In addition, the collection of samples from ileum, cecum and distal colon involved the slaughter of the animals. The complexity of experimental procedures and the ethics committee indications in minimizing the experimental animals limited the number of pigs. Moreover, the scarce interaction effects among the main factors would indicate that, in this scenario, experimental underpowering was not a relevant issue, and hence the number of animals used was considered adequate to achieve the proposed objectives.

### 4.1. Effect of Production Phase

Differences in intestinal bacterial composition between the two PPs were studied since animal age regulates changes in nutrient digestibility and metabolism, immunity and hormone status, and tissue development with a direct impact on microbiome. Intestinal tract samples from growing and fattening phases were obtained from pigs of two PTs, at about 83 and 154 days of age, respectively. Among all alpha diversity indices analyzed, microbial richness was the unique one that showed significant differences between growing and fattening phases. In all intestinal segments, fattening pigs achieved higher values than growing pigs. This is in agreement with previous reports [22,23], which described increased microbial richness as animal matured. Microbial diversity also increases substantially after weaning as a sign of overall development and stability of microbiome composition [23]. However, several studies indicated that diversity indices reach their highest rates before 150 days of age [22,24,25].

In the present study, the ileum microbiota presented a significant interaction (PP × PT) in their richness. While microbial richness increased with age in F2 pigs, it remained constant between growing and fattening phases in Duroc pigs. In a previous report, ileum microbiota showed high variability in diversity indices across ages [25], suggesting that the upper intestine displays less stability in microbial communities compared to the lower intestine because of the reduced abundance of microorganisms [26]. Authors are not aware whether this variability has a genetic component.

Gut microbiota of both PPs clustered separately in the three intestinal segments in the PLS–DA graphical representations, evidencing a differential microbial composition throughout the productive period, as was previously described [24]. In addition, microbial network complexity in cecum and distal colon increased considerably from growing to fattening phases, especially in the F2 pigs. Similar results were obtained by Ke et al. [22], who found increased interaction network in pigs from 25 to 120 days of age, which may be associated with greater microbial diversity and stability of matured pigs. Higher diversity and microbial network complexity may also lead to improved capacity for digestion and metabolism (e.g., complex carbohydrates, protein) [22]. However, in Duroc pigs the architecture of microbial network remained constant in both PPs. This difference between PTs may be associated with the earlier maturity of castrated Duroc pigs compared to commercial crossbreds [27], along with the earlier development of digestive organs and higher digestive enzyme activity reported in fatty breeds [28,29].

At the phylum level, although no significant differences were detected in Firmicutes/Bacteroidetes ratio throughout the intestinal tract, it numerically decreased with age. The lower ratio correlates with the increasing proportion of Bacteroidetes phylum as pigs aged found in previous studies [12,30].

### 4.2. Effect of Producing Type of Pig

The study of gut microbiota between the two extreme PTs of pig aimed to elucidate certain microbiota traits responsible for the phenotypic differences. Duroc pig is a preferred breed for producing premium dry-cured products due to its high intramuscular fat content, and excellent fatty acid composition [31]. To promote such features, pigs are sacrificed at high weights and, therefore, males are castrated to avoid boar taint. On the other hand, F2 pigs were selected to improve productive parameters in order to obtain low-priced lean meat, thus, pigs are slaughtered earlier and are not castrated. Since the effect of sex and breed are not separated, the definition of breed was replaced by the term “producing type” that also includes the effect of castration in the case of Duroc males.

PT also impacted the alpha diversity, with F2 pigs having significantly higher microbial diversity than Duroc pigs across the three intestinal segments, according to Shannon and Simpson indices. Moreover, F2 pigs also presented higher microbial richness (in cecum) and evenness (in ileum and colon). This implies that F2 pigs host a higher number of different taxa with more similar abundances. Considering that the rearing conditions throughout the experiment were controlled, these differences may be correlated with physiological traits. These results are in agreement with those obtained by several studies [32,33,34] in which the leaner pig breeds presented higher diversity indices than fattier or unimproved ones. For instance, Duroc showed lower diversity than Landrace pure breed [32,33], Landrace being the typical genetically lean pig. However, the opposite was also reported, with Duroc and Jinhua breeds having higher diversity than Landrace, Hampshire and Yorkshire [35,36]. Discrepancies between these studies may have resulted from different diet compositions, environmental conditions or the use of distinct intestinal segment contents. Higher bacterial diversity is generally considered favorable in terms of stability and resilience to dysbiosis and potential pathogen threats [37]. In addition, F2 pigs presented a more complex microbial network architecture in terms of both node degree and betweenness centrality, which may also contribute to their higher robustness [38], that is the microbial community’s ability to cope with disturbances [39]. These traits may be responsible for their improved indices of producing performance and apparent CP digestibility obtained in a previous trial [34]. Authors are not aware of the interaction effect of breed and sex on microbial populations. Contrary to our results, Wang et al. [40] reported that castrated Hainan special wild boars had significantly higher diversity than entire ones, caused by their decreased androgen secretion [41]. Our results may suggest that genetic background has a more important influence on the microbial composition than sex, as was previously described [42].

Regarding Firmicutes/Bacteroidetes ratio, the balance between these two dominant phylogenetic types in the gut microbiota has been closely related to adiposity and fat metabolism [6]. However, this ratio varies across intestinal segments and over time due to dynamic compositional changes. In the present study, Duroc pigs tended to have a higher ratio in the cecum than F2 pigs, which is consistent with the results obtained by Guo et al. [43] who evidenced that the percentage of Bacteroidetes in the cecum segment had a negative correlation with backfat thickness (R^2^ = 0.63). Although no significant differences were found in their study, Firmicutes phylum was numerically higher in obese pigs [43], suggesting the potential implication of such phylum in carbohydrates degradation and subsequent fat deposition. In the case of distal colon, no significant differences were detected between PTs and PPs. However, the Firmicutes/Bacteroidetes ratio of growing Duroc pigs was numerically higher than fattening Duroc pigs and F2 pigs in both PPs. Despite the limitations of comparing different studies, the increased proportion of Bacteroidetes with age in Duroc pigs may be in accordance with Crespo-Piazuelo et al. [26] who found that Iberian pigs of almost 50 kg BW had a considerable proportion of Bacteroidetes in the colon segment. In addition, feces from 240-day-old Jinhua fatty pigs also showed higher relative abundance of Bacteroidetes than Landrace pigs [6].

Microbial community in ileum and cecum segments clustered separately between both PTs of pig through beta diversity analyses, although no genera with significantly different abundance could be identified. However, numerical differences in genera abundance between fatty and lean pigs are in accordance with previous studies. Other authors have already found slightly higher levels of *Lactobacillus* genus in high quality meat breeds [36,44]. Moreover, several studies also observed that *Clostridium* is found in higher proportion in fattier pigs than in their lean counterparts [9,33]. Duroc pigs also harbored higher abundance of *Streptococcus* and *Bifidobacterium* genera, which were also higher in Jinhua pigs [35] than in Landrace.

### 4.3. Effect of Dietary CP Content

Although breeding selection is generating highly efficient animals, there is still some questions over their gut microbiota adaptation to nitrogen-restricted diets. Two percentage units were reduced in the experimental LP diets with respect to SP. All diets were supplemented with essential amino acids to meet the nutritional requirements for both growing and fattening phases [14]. This approach is an established strategy in the precision feeding systems. The objective is to reduce the nitrogen load from manure, which has been demonstrated to improve nitrogen utilization without compromising pig performance indices [34,45], and to protect animals against intestinal disorders [3,46]. In addition, the 2% CP difference between SP and LP diets has resulted in similar differences in ether extract content, which derives from maintaining a comparable metabolizable energy value (Table 1). Although the latter difference may have some degree of impact on hindgut microbiota [47], priority was given to maintaining metabolizable energy levels.

Although gut microbiota diversity was not affected during the 15-day experiment, LP-fed pigs presented more complex microbial networks than SP-fed pigs along the three intestinal segments, especially in ileum and colon segments. In the cecum, this effect was registered only during the growing period. These results suggested that, although the limited impact of a 2% CP restriction, microbiota underwent a controlled adaptation process, revealing new relationships between microbes that may result in new metabolic pathways. A similar phenomenon was previously described in ruminants [48] and malnourished children [49], leading to improved nutrient utilization and maintenance of normal physiology. In accordance with our results, neither Zhou et al. [50] nor Seradj et al. [34] found variation in microbial diversity with the use of moderate change in CP (2–3%) level, whereas a slight reduction in the abundance of certain genera was found in their low protein diet. These previous experiments justified their results by a long-term adaptation of the gut microbiota to nutrient availability. However, the present study only lasted for 15 days, suggesting rapid adaptation.

### 4.4. Microbiota Composition throughout the Intestinal Tract

Several studies have described the variation in microbial populations across the intestinal segments, which are anatomically and functionally distinct. In addition, microbial populations are modified by inherent variations in the environmental conditions (i.e., pH, molecular oxygen and oxidation/reduction potential), transit time, substrate and the presence of gut receptors throughout the intestinal tract [12]. The more neutral pH, slower time of transit and higher substrate availability in the more distal intestinal segments [12,51] allow a higher microbial growth and diversity [26,52]. Therefore, the highest number of OTUs in this study was found in the distal colon (453 OTUs), followed by the cecum (395 OTUs) and the ileum (181 OTUs). Similarly, diversity indices followed the same progression, being higher in the lower intestinal segments than in the ileum. In addition, the increased microbial interactions reported in the present study in the lower gut segments may be coincident with the improved microbial communities stability [51].

In addition, microbial composition is subjected to other factors such as genetics, age, diet and environmental conditions, therefore, it is difficult to compare the ratio of species abundance between different studies [12]. Firmicutes/Bacteroidetes ratio in ileum could not be calculated due to the low proportion of Bacteroidetes phylum in this gut segment, which is consistent with the existing literature [53]. Recent studies [25,52] showed that microbial composition of the upper and lower intestines are different. While the upper intestine (including ileum) harbors a higher abundance of Firmicutes and Proteobacteria phyla, the lower intestine (including cecum and distal colon) contains a higher proportion of Bacteroidetes, although Firmicutes remains the most abundant phylum in this segment too [26,52]. The lower Firmicutes/Bacteroidetes ratio of the distal colon compared to the cecum may be due to the increased proportion of Bacteroidetes and the decreased proportion of Firmicutes from proximal to distal parts [12,52]. In contrast, the opposite relationship was also reported by previous authors [53].

As was previously defined [25], the most abundant OTUs found throughout the intestinal tract belonged to the *Lactobacillus* genus. *Lactobacillus* was also the most predominant genus of the core microbiota of all intestinal segments [22,51], with the exception of distal colon of growing pigs. This genus has been related with immunological response of the host. Clostridia class (*Terrisporobacter* and *Clostridium*) was also abundant, especially in the fattening phase, and showed similar pattern of change across intestinal segments [50,51]. *Bifidobacterium* was more abundant in the upper intestine [52], while Ruminococcaceae and Lachnospiraceae families were more abundant in the lower intestine [13,52]. The latter has been associated with degradation and fermentation of carbohydrates and subsequent production of short-chain fatty acids, such as butyric acid [51].

## 5. Conclusions

The microbial community was mainly affected by the PP and PT of pig. Fattening pigs showed higher microbial richness than growing pigs, which is attributable to the higher maturity and stability of their microbial community. In addition, F2 pigs presented higher bacterial diversity and microbial network complexity, especially in the fattening phase, while Duroc pigs tended to have higher Firmicutes/Bacteroidetes ratio in cecum. The moderate restriction in dietary CP content increased the complexity of microbial interaction network, whereas it had limited impact on microbial community composition.

## Figures and Tables

**Figure 1 animals-11-02846-f001:**
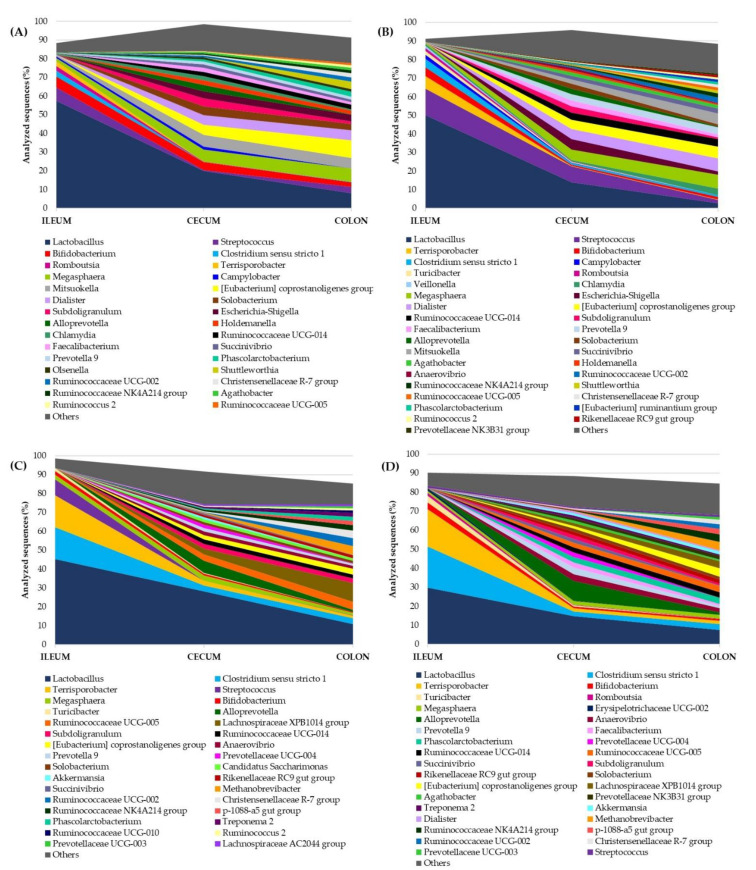
Genera abundance throughout the intestinal tract in (**A**) growing Duroc pigs (26.4 kg), (**B**) growing F2 pigs (31.6 kg), (**C**) fattening Duroc pigs (85.1 kg) and (**D**) fattening F2 pigs (91.1 kg). Low-abundance genera (<1% analyzed sequences) are represented as “Others”.

**Figure 2 animals-11-02846-f002:**
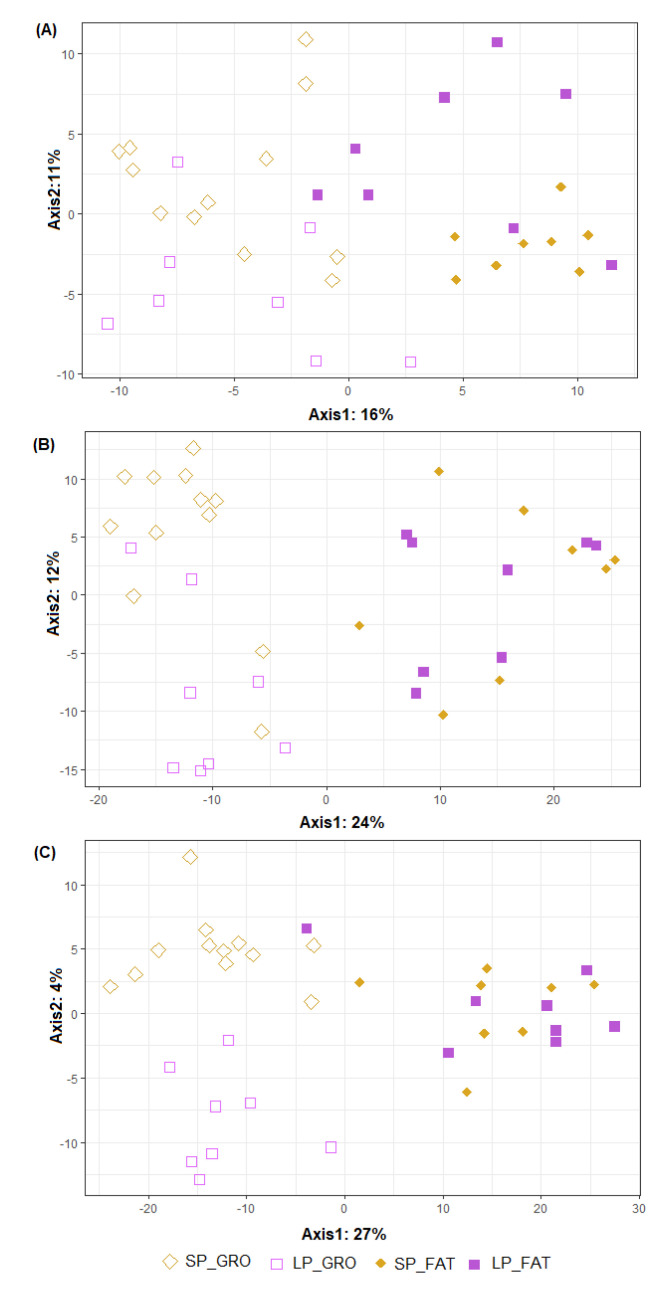
Graphical representation of partial least squares-discriminant analysis (PLS−DA) on microbial OTUs in ileum (**A**), cecum (**B**) and distal colon (**C**). Obtained in pigs differing in their protein intake: standard protein (SP) vs. low protein (LP) and in their production phase: growing (GRO: 28.5 kg) vs. fattening (FAT: 88.1 kg). Each point represents a different sample and a greater distance between two points infers a higher dissimilarity between them. Statistical comparisons were made between PP (*p* < 0.001 for all intestinal segments), between diets (*p* = 0.916, 0.433 and 0.573 for ileum, cecum and distal colon, respectively) and to test PP by diet interaction (*p* = 0.239, 0.489 and 0.638 for ileum, cecum and distal colon, respectively).

**Figure 3 animals-11-02846-f003:**
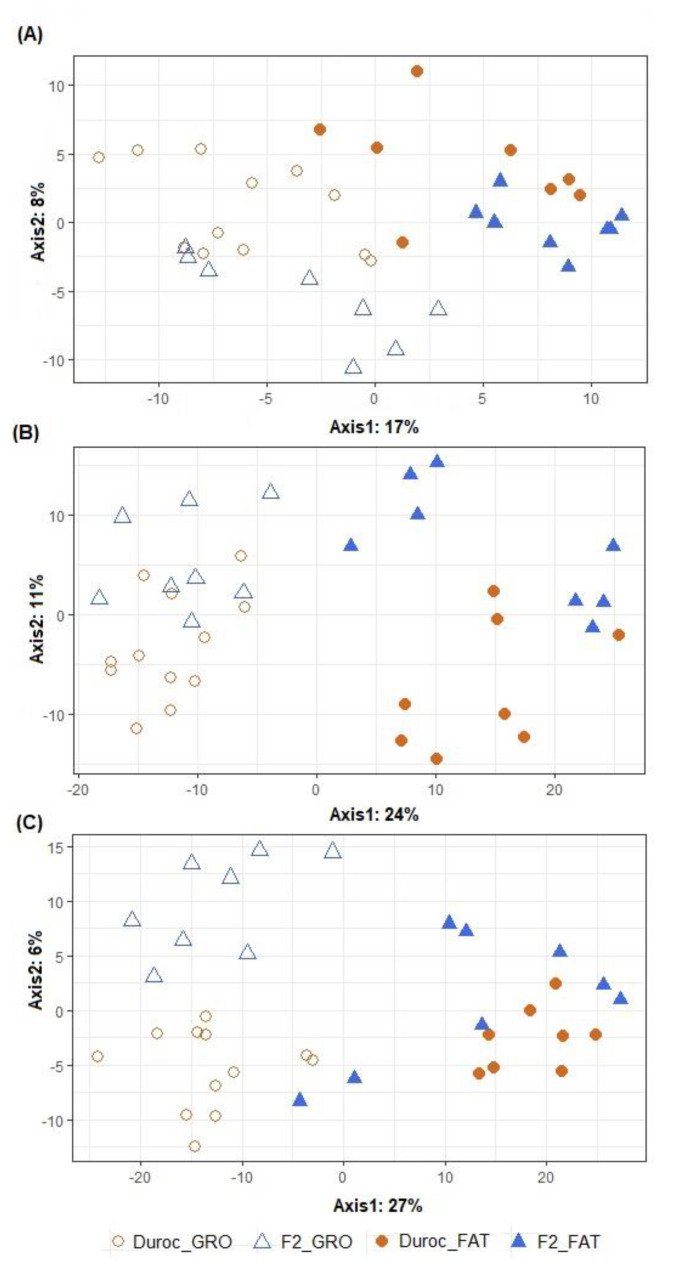
Graphical representation of partial least squares-discriminant analysis (PLS–DA) on microbial OTUs in ileum (**A**), cecum (**B**) and distal colon (**C**). Obtained in pigs differing in their producing type (PT): Duroc vs. F2 (Pietrain × F1: Duroc × Landrace) and production phase (PP): growing (GRO: 28.5 kg) vs. fattening (FAT: 88.1 kg). Each point represents a different sample and a greater distance between two points infers a higher dissimilarity between them. Statistical comparisons were made between PP (*p* < 0.001 for all intestinal segments), between PT (*p* = 0.044, 0.032 and 0.171 for ileum, cecum and distal colon, respectively) and to test PP by PT interaction (*p* = 0.667, 0.542 and 0.232 for ileum, cecum and distal colon, respectively).

**Table 1 animals-11-02846-t001:** Ingredients and chemical composition of the two-phase experimental diets.

Items ^1^	Growing Phase		Fattening Phase	
	Low Protein	Standard Protein	Low Protein	Standard Protein
Ingredients (g/kg DM)				
Corn	294.8	246.5	190.3	99.2
Barley	290.0	287.8	466.0	512.8
Wheat	200.0	200.0	200.0	200.0
Soybean meal	137.6	195.0	65.8	113.6
Beet pulp dehydrated	30.0	30.0	30.0	30.0
Calcium carbonate	13.4	13.2	10.0	10.0
Mono calcium phosphate	9.4	8.8	7.0	6.4
Soybean oil	9.0	6.3	17.0	17.1
L-Valine	6.8	8.0	0.0	0.0
Sodium chloride	4.6	4.6	4.1	4.1
L-Lysine HCL	4.2	2.4	3.7	2.1
Vitamin-Mineral mix ^2^	4.0	4.0	4.0	4.0
L-Threonine	1.6	0.8	1.2	0.5
DL- Methionine	1.0	0.5	0.6	0.3
L-Tryptophan	0.3	0.2	0.2	0.0
Chemical composition (g/kg DM)				
Dry matter (g/kg FM)	900.1	897.1	909.9	908.4
Crude protein	147.2	166.9	129.27	154.95
Ether extract	24.9	26.9	39.7	43.2
Ash	65.8	70.3	58.1	52.8
NDF	173.3	179.9	211.8	211.6
ME (kcal/kg)	3108	3094	3185	3185

^1^ Complete diets differed in 2% crude protein concentration (standard protein vs. low protein) for pigs in growing and fattening production phases. DM: dry matter; FM: fresh matter; ME: metabolizable energy, NDF: neutral detergent fiber. ^2^ The vitamin and mineral mixture composition is available in Sarri et al. [15].

**Table 2 animals-11-02846-t002:** Microbial alpha diversity indices (based on OTUs), and Firmicutes/Bacteroidetes ratio (ratio F/B) in ileum, cecum and distal colon segments.

Item ^1^	Growing Phase		Fattening Phase		SEM	*p*-Value	
	Duroc	F2	Duroc	F2		PT	PP
**n**	12	8	8	8			
Ileum ^2^							
Shannon index	1.53 b	1.64 ab	1.47 b	2.18 a	0.160	0.014	0.186
Simpson index	0.61 b	0.69 ab	0.66 ab	0.81 a	0.049	0.009	0.102
Richness ^3^	31.48 ab	24.38 b	31.50 ab	43.88 a	3.854	0.446	0.020
Evenness	0.45	0.52	0.43	0.58	0.040	0.011	0.785
Cecum							
Shannon index	3.18	3.35	3.22	3.76	0.177	0.033	0.189
Simpson index	0.92	0.92	0.91	0.95	0.018	0.422	0.618
Richness	90.23 b	104.88 ab	106.50 ab	134.50 a	8.973	0.010	0.013
Evenness	0.71	0.72	0.70	0.77	0.034	0.200	0.457
Ratio F/B	47.57	24.40	35.68	2.37	18.409	0.089	0.309
Distal colon							
Shannon index	3.53	3.83	3.67	3.77	0.128	0.076	0.650
Simpson index	0.94	0.96	0.94	0.95	0.009	0.047	0.742
Richness	123.64	140.00	146.13	153.72	9.595	0.138	0.045
Evenness	0.74	0.78	0.74	0.78	0.017	0.011	0.820
Ratio F/B	28.73	10.92	7.65	11.49	8.110	0.185	0.112

^1^ Obtained in pigs differing in their producing type (PT): purebred Duroc vs. crossbred F2 (Pietrain × F1: Duroc × Landrace) and production phase (PP): growing (28.5 kg) vs. fattening (88.1 kg). Standard error of the mean (SEM) and significance of PT and PP effects are shown. Mean values within a row followed by different letters differ significantly at *p* = 0.05. ^2^ Ratio F/B could not be calculated in ileum samples due to the low abundance of Bacteroidetes phylum in that intestinal tract (0.28% of analyzed sequences). ^3^ Only one PT by PP interaction was significant (ileum-richness: *p* = 0.014) and the rest of the interactions were not included in the table.

## Data Availability

The data presented in this study are available on request from the corresponding author.

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
