# Peer review of "The Impact of Genetics on Gut Microbiota of Growing and Fattening Pigs under Moderate N Restriction"

_animals, 2021, doi:10.3390/ani11102846_

Round 1

Reviewer 1 Report

see document attached

Author Response

Response to Reviewer 1 Comments

Point 1. L152: 36 (animals) x 3 (3 intestinal sections) = 108 digestive content samples?

Authors agree that the way it was written could lead to confusion. Text in L159 has been amended to clarify it. The number of samples is 108 (36 animals x 3 sections)

Point 2. L123-125: I do not quite understand why this specific statistical test was used. Should not it be a factorial design, i.e., 2 (growing phase) x 2 (producing type) x 3 (intestinal section)?

We applied the following linear model on the OTUs dataset of each intestinal section.

            Alpha diversity metrics ~ PT*PP + Diet*PP + e

We avoided to create a single dataset including the three intestinal sections studied in order to decrease datasets sparsity: certain OTUs are not present in each one of the three locations and, therefore, that would highly increase the number of zeros.

Point 3. Table 1: Differences in ether extract values between diets are 2-3.5%, similar to differences in protein values. Would this have an effect on intestinal microbiota composition?

In order to study the effect of a restriction in crude protein content on microbial populations, the SP and LP diets were formulated to keep the 2% difference, but meeting the same nutritional requirements recommended by the Spanish Foundation for the Development of Animal Nutrition (FEDNA). This has resulted in differences in the ether extract content to maintain an almost equal metabolizable energy value (Table 1).

These differences in ether extract may have a limited impact on microbial populations, since it is a highly digestible nutrient that is absorbed very efficiently in the small intestine. Moreover, fat is a nutrient that bacteria generally cannot degrade. Thus, we believe that although 2% difference in EE may have some degree of impact on hindgut microbiota, it was better to apply this difference rather than use diet with differences in metabolizable energy. There is an interesting area of study in the interaction energy:protein, but it is out of the scope of this study.

Point 4. Page 5, Table 2: At what level (phylum, class, order, etc.) were the analysis carried out? Are significant differences expressed within each phase or for both phases? P values for Evenness in ileum and distal colon and Simpson index in distal colon were < 0.05 for PT. Why that was not indicated with superscript letters?

Alpha diversity was calculated based on OTUs, authors have indicated it in the heading of the Table 2.

Significant differences are expressed for both phases.

P values showed in Table 2 indicate the significance of the effects PT and PP in the linear model by an ANOVA test and, in turn, superscript letters indicate the results of the multiple comparisons between means by a Tukey test. In some cases, due to the limited number of animals, the global effect can be significant but, when making pairwise comparisons between the particular means, that may not reach statistical significance.

Point 5. L191-193: In absence of proper statistical comparisons between segments, observations for this issue should be taken as provisional (lines 191-198) and conclusions considered with caution. Why no statistical comparisons were made between the three intestinal segments?

We have reinforced the warning to take the comparisons among the three intestinal segments with caution (L198-199).

To compare between the three intestinal segments was not one of the objectives of the present study, since the extra data would divert the reader’s attention from the main purpose without adding any relevant information. Differences in microbial populations through the intestine have been extensively studied recently by several authors such as Zhao et al. (2015), De Rodas et al. (2018), Crespo-Piazuelo et al. (2018), Gresse et al. (2019) and Wang et al. (2019).

Point 6. L194-196: This is not true in the figure for all genera and evenness was not numerically higher in Table 2 in the lower intestine.

The authors disagree with this comment. Although no statistical analysis was performed among intestinal segments, the number of major genera (>1% analyzed sequences) in the lower intestine, including cecum and distal colon, was greater than in the upper intestine (ileum), which is visually evident in the four figures 2A, 2B, 2C and 2D. Specifically, the ileum segment contained a mean number of 15 major genera in all groups, whereas the lower intestine segments harbored a mean number of 32 major genera. In addition, evenness was numerically higher in the lower intestine (cecum and distal colon) compared to the ileum, as documented in Table 2: while the evenness in the ileum was about 0.50, in the lower intestine it was above 0.70.

Point 7. Page 7, Figure 1: In absence of proper statistical comparisons between segments this figure can be regarded only as illustrative and should probably be reported as supplementary material.

Point 8. L223-224: I think this figure is more relevant than Figure 1 and should probably be in the main text. Where are the statistical results for this discriminant analysis reported? They should be in the footnote.

From the author’s point of view, Supplementary Figure S4 is less relevant than the present Figure 1 since no significant differences in beta diversity were reported between the two experimental diets SP and LP. On the other hand, although significant differences between production phases were evidenced in the three intestinal segments, the same differences are already represented in the present Figure 2. In addition, Figure 1 allows a comparison of how major genera were distributed along the three intestinal segments between both producing types and production phases.  Anyway, if reviewer maintains his opinion in this respect, we can accept to include Figure S4 in the main text in further rounds of revision.

Authors have included the statistical results of the discriminant analysis in the footnote of Figure S4.

Point 9. L229-230: Where is the comparison between ileum, cecum and distal colon?

Authors have included the statistical results of the discriminant analysis in the footnote of Figure 2.

Point 10. Page 8, Figure 2: Where are the statistical results for this discriminant analysis reported? They should be in the footnote.

Authors have included the statistical results of the discriminant analysis in the footnote of Figure 2.

Point 11. L246-248: Where do these figures come from?

Network analysis is described in the section 2.3 of Materials and methods (L147-155); the results derived from this analysis are shown in Supplementary Table S2, and Supplementary figures S5-S10 are the graphical representation of network metrics of Table S2, as commented in the main text (L253-256).

Point 12. L248-259: Could this complexity in networks be quantified in any way?

Network analysis is described in the section 2.3 of Materials and methods (L147-155) and network metrics are shown in the Supplementary Table S2. Authors have indicated where these results come from in the main text (L259-262).

Point 13. L260-266: Can this betweenness in networks be quantified in any way?

Network metrics are shown in the Supplementary Table S2 as was commented in the main text (L272).

Point 14. L278-279: Which were pigs of identical populations?

Authors have changed the term “identical populations” for “two producing types” (L292-293)

Point 15. L279-281: What about richness in the ileum?

Richness in the ileum got also significant differences between both production phases (P = 0.020; Table 2)

Point 16. L287-288: Where is the value of this interaction reported?

The value of the interaction (PP x PT; P = 0.014) is reported in section 3.2 of Results (L179), and also in the footnote of Table 2.

Point 17. L371: 4.3

We have corrected the number of the section 4.3 in Discussion

Point 18. L372-373: there are still some questions?

We have changed “answers” to “questions” in L388.

Point 19. L381: How was this quantified?

Network metrics are shown in the Supplementary Table S2.

Reviewer 2 Report

I found this manuscript interesting and with great potential for readers. The main part of the study - assessment of the microbiome (in Methods and Results description) -  is well presented and described.

I have some objections to the first part of Methods - animals study design,  and later to some parts of discuss.

Describe more detailed the treatment of animals. I am confused after your presentation. At begin we can find that pure breed Duroc pigs as well as crossbreed F2 were used in the study and there is presented mean body weight 26,4 and 86,1 for Duroc and 30,5 and 91,1 for F2 - what is it? is it a body weight of slaughtered pigs in both PT and PP? We can find that every feed mixture was given ad libitum for 11 days? Between the first and second (final?) BW we have more than 60 kg differences so what to think about these 11 days? I suppose that probably after weaning you start to use the first mixture, the finish was when animals obtained about 26 or 30 kg and some of them were slaughtered, next there was use the second mixture as long as animals obtained about 88 or 90 kg - is it true? Animals design is very poor and you have to correct it. That's fine that animals were in the same environment but you have to describe it more.

I agree that Iberian or Jinhua pigs exibit propensity to obesity but Duroc not necessary. Characteristic for Duroc is ability to store intramuscular fat (thanks to it their meat is more tasted) but we shouldn't write that these pigs belong to fattier breeds. In your experimental scheme we have two factors: PP and PT. Production phase (PP) is correct and we can use it, but production type (PT) - this term can be confusing. I don't see too many differences between used crossbreed pigs and Duroc - it is the same meat type. Of course it is your description used in the manuscript (so it could be correct), but from the discuss I see that it was wrong decision. For this study the second factor should be described as G (genotype or genetic) it will be more precise because we have pigs in the same type. Usually we can expect earlier maturity and more rapid growth in crossbreed pigs thanks to phenomenon of heterosis, so your explanation from 303-305 line is not correct and you can't confirm this sentence using Hutchens et al. (1982) paper. I don't agree that we can replaced definition of breed by the term "producing type" - it is an abuse. You should not write that Duroc pigs are obese.

I think this paper is very good, we can find here interesting conclusions. I recommend strongly to introduce some mentioned corrections and good luck ;).

Author Response

Response to Reviewer 1 Comments

Point 1. Describe more detailed the treatment of animals. I am confused after your presentation. At begin we can find that pure breed Duroc pigs as well as crossbreed F2 were used in the study and there is presented mean body weight 26,4 and 86,1 for Duroc and 30,5 and 91,1 for F2 - what is it? is it a body weight of slaughtered pigs in both PT and PP?

The weight detailed in the Material and methods section is the weight of slaughtered pigs; it is mentioned in the text (L79).

Point 2. We can find that every feed mixture was given ad libitum for 11 days? Between the first and second (final?) BW we have more than 60 kg differences so what to think about these 11 days? I suppose that probably after weaning you start to use the first mixture, the finish was when animals obtained about 26 or 30 kg and some of them were slaughtered, next there was use the second mixture as long as animals obtained about 88 or 90 kg - is it true? Animals design is very poor and you have to correct it.

Authors agree with referee’s point of view with the need to improve the description of the experimental design. Adaptation to the new experimental diets consisted in 4 days of dietary changeover, from the commercial to the experimental diet, plus 11 days of adaptation to the experimental ration.

As will be later developed, the aim of the authors was to perform a complete comparison among metabolic status between both “producing types” including protein turnover and fat metabolism and to use the same individual was considered as a critical point for the authors. The complete experimental design was a framework for the actual experimental design. In any case the mentioned section has been amended in order to increase clarity (75-89).

Point 3. That's fine that animals were in the same environment, but you have to describe it more.

Following reviewer’s suggestion, authors have introduced a brief description of the environmental conditions (L90-95).

Point 4. I agree that Iberian or Jinhua pigs exhibit propensity to obesity but Duroc not necessary. Characteristic for Duroc is ability to store intramuscular fat (thanks to it their meat is more tasted) but we shouldn't write that these pigs belong to fattier breeds. In your experimental scheme we have two factors: PP and PT. Production phase (PP) is correct, and we can use it, but production type (PT) - this term can be confusing. I don't see too many differences between used crossbreed pigs and Duroc - it is the same meat type. Of course, it is your description used in the manuscript (so it could be correct), but from the discuss I see that it was wrong decision. For this study the second factor should be described as G (genotype or genetic) it will be more precise because we have pigs in the same type. Usually, we can expect earlier maturity and more rapid growth in crossbreed pigs thanks to phenomenon of heterosis, so your explanation from 303-305 line is not correct and you can't confirm this sentence using Hutchens et al. (1982) paper. I don't agree that we can replaced definition of breed by the term "producing type" - it is an abuse. You should not write that Duroc pigs are obese.

The aim of the present study was to determine the differences between two types of pigs used commercially for different production purposes, or rather, to obtain two different meat products. F2 crossbred pigs are used to produce lean carcasses and are slaughtered at lighter weights to obtain the optimum growth potential and feed efficiency, using entire males. However, Duroc pigs, used commercially for dry-cured ham and other high-value meat pieces are slaughtered at heavier weights to improve meat quality and obtain larger pieces, thus, male pigs are usually castrated to avoid boar taint. For this reason, it is necessary to employ the term “producing type”, to reflect differences in both genotype but also sex condition.

In fact, pure Duroc pigs are usually used to obtain specific meat pieces (i.e. dry ham) but beside crossed with Iberian pig to improve growth rate and feed efficiency of the latter without decreasing the meat quality (mostly in Southern Europe).

Probably referee is right and the term “obese” for Duroc pigs may be inappropriate, for this reason, authors have changed “fatty” to “heavy pig” in the text.

Reviewer 3 Report

Manuscript animals-1385261, entitled “The Impact of Genetics on Gut Microbiota of Growing and Fattening Pigs Under Moderate N Restriction”

Recommendation:       The above paper is not suitable for publication in its present form.

General comment

The article provides useful information about the effects of “producing type”, producing phase and crude protein levels on gut microbiota. Although, the experiment is in general appropriately designed and implemented, there are some points that should be corrected or clarified.

Major comments

  • My main concern is the small sample size. Only 4 animals per group (with the exception of Duroc in growing phase that were 6). Please provide a Power Analysis
  • Was the duration of the experiment (11 days) sufficient for showing CP level effects? Authors concluded that CP restriction had moderate – low effects. Can the short experimental duration contribute to these non-significant effects?
  • How were the animals fed before the experimental period? CP levels of the diets?
  • Fattening pigs presented more complex microbial network than growing pigs (L27-28, L432) or growing pigs showed more complexity in their microbial networks than fattening pigs (L250-251)? Please clarify
  • In the section of “Statistical analysis” please add a sentence referring to the interactions of PP by PT or CP.

Minor points

L19: “…the optimization of pork production efficiency.”

L41-42: “…allows the improvement of nitrogen utilization and the reduction of the nitrogen…”

L43: “the” instead of “about”

L44: “palatability” instead of “eating quality”

L45: “…development of alternative production systems of…”

L55: Please delete “the”

L58-59: Please rephrase

L88: Please check reference style of the journal

L121-123: Please rephrase

L170-171: Concerning richness, P-value for PT in ileum and distal colon ia 0.446 and 0.138, respectively. Please check and rephrase

L177: “PP” instead of “age”

L191: “among” instead of “between”

L198: What do you mean by “throughout the intestine”? From ileum to distal colon?

L249: “observed” instead of “seen”

L268-273: Please delete

L289: This study does not refer to PT but to PP. Please delete or rephrase

L306: “…activity reported in fatty breeds.”

L329: “pure breed” instead of “purebred”

L339: “improved” instead of “better”

L343-344: I think that this conclusion is not completely correct, since authors examine “producing type” that includes breed but partially also sex (castration).

L358: May be in line?

L379: “…and to protect animals against intestinal disorders [3,45].”

L405: “reported” instead of “seen”

L407-409: Please remove “[12]” at the end of the sentence.

Author Response

Response to Reviewer 3 Comments

Point 1. The article provides useful information about the effects of “producing type”, producing phase and crude protein levels on gut microbiota. Although, the experiment is in general appropriately designed and implemented, there are some points that should be corrected or clarified.

Major comments: 

My main concern is the small sample size. Only 4 animals per group (with the exception of Duroc in growing phase that were 6). Please provide a Power Analysis

Authors agree with the referee’s point of view regarding the small sample size. The present study is part of a complex trial in which, apart from microbiota, details of animal’s protein and lipid metabolism, including the use of markers, were also studied. In addition, the collection of samples from ileum, cecum and distal colon involved the slaughter of the animals. Thus, the complexity of experimental procedures added to the ethics committee instructions in reducing (o minimizing) the experimental animals set up were the reasons for this constraint. On the other hand, scarce 2-way interactions were found, thus we consider that number of animals was sufficient to assess main factors statistical significance. This has been included in discussion section (L279-287)

Previous studies also have been used similar animal sample size, such as:

  • Yang H., Yang H., Xiang Y., Robinson K., Wang J., Zhang G., Zhao J., Xiao Y., 2018. Gut microbiota is a major contributor to adiposity in pigs. Frontiers in Microbiology 9, 1–13

In where they used 10 Jinhua and 10 Landrace pigs, 5 males and 5 females in each breed.

  • Guo X., Xia X., Tang R., Wang K., 2008. Real-time PCR quantification of the predominant bacterial divisions in the distal gut of Meishan and Landrace pigs. Anaerobe 14, 224–228. https://doi.org/10.1016/j.anaerobe.2008.04.001

In where they used 6 Landrace and 6 Meishan pigs.

  • Xiao Y., Li K., Xiang Y., Zhou W., Gui G., Yang H., 2017. The fecal microbiota composition of boar Duroc, Yorkshire, Landrace and Hampshire pigs. Asian-Australasian Journal of Animal Sciences 30, 1456–1463. https://doi.org/10.5713/ajas.16.0746

In where they used 6 Duroc, 4 Hampshire, 7 Landrace and 6 Yorkshire pigs.

Point 2. Was the duration of the experiment (11 days) sufficient for showing CP level effects? Authors concluded that CP restriction had moderate – low effects. Can the short experimental duration contribute to these non-significant effects?

According to previous authors such as Zhou et al. (2016) and Seradj et al. (2020), similar changes in crude protein level (2-3%) did not cause significant differences in microbiota diversity even with longer experimental periods. Our results may demonstrate that the gut microbiota adapts to dietary changes faster than previously thought.

Zhou et al. (2016) [http://dx.doi.org/10.1016/j.anaerobe.2015.12.009]

Seradj et al. (2020) [http://dx.doi.org/10.3390/ani10101742]

Point 3. How were the animals fed before the experimental period? CP levels of the diets?

The pigs came from commercial farms in where they were fed an appropriate feed according to their growth stage, which met the nutrient requirements recommended by FEDNA (2013). The CP level in growing and fattening pigs was around 17% and 15%, respectively. 

Point 4. Fattening pigs presented more complex microbial network than growing pigs (L27-28, L432) or growing pigs showed more complexity in their microbial networks than fattening pigs (L250-251)? Please clarify

Growing pigs exhibited greater microbial network complexity in the ileum segment. However, in the cecum and colon segments, network complexity was generally higher in fattening F2 pigs, although in Duroc pigs there were no evident differences between PPs. This description is shown in the section 3.5 of Results (L265-270), moreover, it has been corrected in the Abstract (L27-30) and Conclusions section (L446-452).

Point 5. In the section of “Statistical analysis” please add a sentence referring to the interactions of PP by PT or CP.

The interaction PP x PT and Diet x PP for alpha diversity analysis was detailed in L129. However, authors have included it for beta diversity analysis description (L142).

Minor points:

Point 6. L19: “…the optimization of pork production efficiency.”

The text has been amended.

Point 7. L41-42: “…allows the improvement of nitrogen utilization and the reduction of the nitrogen…”

The text has been amended.

Point 8. L43: “the” instead of “about”.

The text has been amended.

Point 9. L44: “palatability” instead of “eating quality”

The text has been amended.

Point 10. L45: “…development of alternative production systems of…”

The text has been amended.

Point 11. L55: Please delete “the”

The text has been amended.

Point 12. L58-59: Please rephrase

We have improved the sentence.

Point 13. L88: Please check reference style of the journal

The reference style has been revised

Point 14. L121-123: Please rephrase

We have improved the sentence.

Point 15. L170-171: Concerning richness, P-value for PT in ileum and distal colon ia 0.446 and 0.138, respectively. Please check and rephrase

We have rephrased the sentence to avoid confusion.

Point 16. L177: “PP” instead of “age”

The text has been amended.

Point 17. L191: “among” instead of “between”

The text has been amended.

Point 18. L198: What do you mean by “throughout the intestine”? From ileum to distal colon?

We though that “from ileum to distal colon” was better than “throughout the intestine.

Point 19. L249: “observed” instead of “seen”

The text was amended.

Point 20. L268-273: Please delete

The text was deleted.

Point 21. L289: This study does not refer to PT but to PP. Please delete or rephrase

We have rephrased the sentence.

Point 22. L306: “…activity reported in fatty breeds.”

The text has been amended.

Point 23. L329: “pure breed” instead of “purebred”

The text has been amended.

Point 24. L339: “improved” instead of “better”

The text has been amended.

Point 25. L343-344: I think that this conclusion is not completely correct, since authors examine “producing type” that includes breed but partially also sex (castration).

Since the term “producing type” also includes differences in sex status, we found it important to understand whether the sex status also had implications for differences between PTs in alpha diversity indices, apart from genetics. As discussed in the main text, Wang et al. (2020) reported that castrated Hainan special wild boars had significantly higher alpha diversity than entire ones. If this is so, we assume that these sex difference occurs in all pig genetics. Thus, the fact that in our experiment castrated pigs showed less diversity than entire pigs suggests that pig genetics is more responsible for the PT difference.

Point 26. L358: May be in line?

We have replaced this expression with “may be in accordance with”

Point 27. L379: “…and to protect animals against intestinal disorders [3,45].”

The text has been amended.

Point 28. L405: “reported” instead of “seen”

The text has been amended.

Point 29. L407-409: Please remove “[12]” at the end of the sentence.

We have moved the reference at the end of the sentence.

Round 2

Reviewer 1 Report

See document attached

Author Response

Response to Reviewer 1 Comments

Point 1. I think that this explanation is disputable, but in any case, authors should include a comment on it in the text if authors maintain the current analysis.

Authors have included a comment about the statistical model used (L130-133).

Point 2. Please do include a short comment on this in text.

A short comment was included in L423-427.

Point 3. Although I do understand authors point, I do not think that including an analysis on the differences between intestinal sections and a short comment on it would do any harm to the study or the reader. In addition, it would reinforce the results shown in Figure 1.

Following reviewer suggestion, differences in microbial composition between the three intestinal segments have been analyzed in both alpha and beta diversity, and a new Table and Figure have been included in Supplementary material.

Point 4. Yes, I still think that Figure S4 should be in the main text and Figure 1 probably supplemental.

Figure S4 has been included in the main text but we still believe that Figure 1 has the relevance to be kept into the main document, especially after having conducted additional analyses showing those differences between intestinal segments (see Point 3).

Point 5. Could authors include the statistical analysis description in Table S2?

The statistical analysis description was included in the Supplementary Table S2.

Reviewer 3 Report

The manuscript is substantially improved, since authors made all the necessary amendments.  I recommend its acceptance.

Author Response

Thank you for your consideration